# Thrombus characteristics evaluated by acute optical coherence tomography in ST elevation myocardial Infarction

Erlend Eriksen[1,2]*, Jon Herstad[1], Kartika Ratna Pertiwi[3¤], Vegard Tuseth[1,2], Jan Erik Nordrehaug[2], Øyvind Bleie[1☋], Allard C. van der Wal[3☋]

**1** Department of Heart Disease, Haukeland University Hospital, Bergen, Norway, **2** Department of Clinical Science, University of Bergen, Bergen, Norway, **3** Department of Pathology, Amsterdam UMC, University of Amsterdam, Amsterdam, The Netherlands

☋ These authors contributed equally to this work.
¤ Current address: Department of Biology Education, Faculty of Mathematics and Natural Science, Yogyakarta State University, Yogyakarta, Indonesia
* eeer@helse-bergen.no

**Data Availability Statement:** Minimal data set has been uploaded as Supporting Information

**Funding:** The author(s) received no specific funding for this work.

## Abstract

### Aims

ST elevation myocardial infarction (STEMI) is caused by an occlusive thrombosis of a coronary artery. We wanted to assess if the thrombus can be characterized according to erythrocyte content and age using intravascular optical coherence tomography (OCT) in a clinical setting.

### Methods and results

We performed manual thrombus aspiration in 66 STEMI patients. OCT was done of the thrombus remnants after aspiration. A light intensity ratio was measured through the thrombus. Forty two of the aspirates had thrombus which could be analyzed histomorphologically for analysis of erythrocyte and platelet content, and to determine the age of thrombus as fresh, lytic or organized. There were 11 red, 21 white and 10 mixed thrombi. Furthermore, 36 aspirates had elements of fresh, 7 of lytic and 8 of organized thrombi. There was no correlation between colour and age. OCT appearance could not predict erythrocyte or platelet content. The light intensity ratios were not significantly different in fresh, lytic or organized thrombi.

### Conclusion

OCT could not differentiate between red and white thrombi, nor determine thrombus age.

## Introduction

ST-elevation myocardial infarction (STEMI) is a life-threatening situation with mortality of about 10% even after successful primary percutaneous coronary intervention (PPCI) [1]. Time

**Competing interests:** The authors have declared that no competing interests exist.

**Abbreviations:** STEMI, ST elevation Myocardial Infarction; OCT, Optical Coherence Tomography; PPCI, Primary Percutaneous Coronary Intervention; PCI, Percutaneous Coronary Intervention; DAPT, Dual Antiplatelet therapy; GFR, Glomerulus Filtration Rate; ROI, Region of Interest; HE, Hematoxylin and Eosin; EvG, Elastic van Gieson; SMA, Smooth Muscle Actin; SMC, Smooth Muscle Cells; CD, Cluster of Differentiation; NET, Neutrophil Extracellular Trap; MMP, Matrix Metallo Proteinase; HT, Hypertension; DM, Diabetes Mellitus; BP, Blood Pressure; IVUS, Intravascular Ultrasound; AMC, Academisch Medisch Centrum; LAD, Left Anterior Descending artery; RCA, Right Coronary Artery; CX, Circumflex artery; GpIIb/IIIa, Glycoprotein receptor IIb/IIIa.

from onset to revascularization is an important prognostic factor. The underlying cause of the coronary occlusion is often a ruptured lipid rich plaque, leading to thrombus formation and coronary occlusion. In PPCI coronary flow is restored mechanically, and the plaque rupture is sealed with a stent [2, 3].

Identifying the age of the thrombus in real time, could potentially highlight the mechanism of disease, impact treatment and improve outcome.

Previous intracoronary studies have shown that an acute STEMI is often preceded by a longer period of thrombus formation and lysis. More than 50% of the thrombi showed evidence of age > 1 day, with 9% showing evidence of being > 5 days old. Thrombus age of > 1 day predicts worse outcome [4, 5]. The mechanisms have not been evaluated.

A previous post mortem ex vivo study has demonstrated that thrombus colour, a macroscopic evaluation of erythrocyte content, can be predicted by Optical Coherence Tomography (OCT). The study compared the macroscopic colour of thrombi with the OCT image in a standardized ex vivo model [6, 7].

Our study is a substudy of "Perfomance of Bioresorbable Scaffold in Primary Percutaneous Intervention of ST Elevation Myocardial Infarction". Manuscript of this study has been submitted. We wanted to assess if OCT can predict the erythrocyte content and age of thrombi aspirated from STEMI patient.

## Methods

### Aim of study

The primary endpoint was the loss of backscattered light through the coronary thrombi measured as a light intensity ratio by OCT. This enable us to differentiate between erythrocyte rich thrombi (red) and platelet rich thrombi (white) [7]. The relative distribution of thrombocytes, erythrocytes and fibrin changes with ischemic time [8]. We therefore hypothesize that OCT can differentiate between fresh, lytic and organized thrombi. This could have prognostic and treatment relevance.

### Patient selection

All patients >18 years presenting for PPCI with STEMI were eligible for inclusion. Important exclusion criteria were contraindications to long-term dual antiplatelet therapy (DAPT), known renal failure (Glomerulus filtration rate (GFR) < 45), cardiac arrest or persistent cardiogenic shock. Procedural contraindications were severely calcified or tortuous vessel, large side branch ($\geq$2.5 mm) at culprit lesion or unable to advance aspiration catheter past the occlusion. The study was conducted in accordance with the protocol, applicable regulatory requirements and the ethical principles of the Declaration of Helsinki as adopted by the 18th World Medical Assembly in Helsinki, Finland in 1964 and subsequent versions. The study was approved by the Regional Ethics Committee of Norway, Region West (2013/2006) and registered at www. clinicaltrials.gov (NCT02067091). All patients signed a written informed consent.

### Collection of thrombus

After identifying culprit lesion in a coronary artery by angiography, collection of thrombus was performed with a manual aspiration catheter (Export advance©, Medtronic, Minnesota, USA) passed through the lesion. The aspirate was filtered through a 40 micron filter basket and fixated in 4% buffered formalin for at least 24 hours before shipment. Further investigation of the thrombus was done at the Department of Pathology laboratory, Academisch Medisch Centrum (AMC), University of Amsterdam, The Netherlands.

## Imaging

Ilumien Optis© (Abbott, Illinois, USA) OCT system was used. The Dragonfly© (Abbott, Illinois, USA) catheter was introduced into the culprit vessel after restoring flow with thrombus aspiration catheter. Further balloon dilatation before imaging was discouraged, but sometimes necessary to achieve images. Images were taken in high-resolution with pullback speed of 20 mm/s, using automatic trigger for the first run (automatic injection speed 4 ml/s, max 18 ml). If more runs were needed, adjustments were made at operators discretion. Thrombi were identified in the stored recordings in co-operation with the OCT core lab at Aarhus University Hospital, Skejby, Denmark.

## OCT Ilumien Optis software

The OCT images were analyzed using Ilumien Optis© software. Manual calibration was performed. Thrombus was defined as irregular intraluminar structure visible over several consecutive images. An area of definite thrombus > 0,25 mm in depth was identified in the pre percutaneous coronary intervention (PCI) images. If this was missing or of poor quality, post PCI images were scrutinized for thrombus remnants. If more than one area of thrombus were identified, the one with least backscattering (i.e. most "red" looking) was selected. Images were analyzed using 2x zoom. Thrombus not immediately adjacent to the catheter was preferred. The 0 mm region of interest (ROI) was marked at the luminal edge. A 0,25 mm marker was measured in an axial direction from center. The 0,25 mm ROI was then marked immediately distal to the 0,25 mm mark. A ratio of light intensity loss through the thrombus was then calculated. Two investigators at Haukeland University Hospital performed the measurements individually.

## OCT ImageJ software

Raw data from the OCT recording was exported to an off-line station with ImageJ© (National Institute of Health and the Laboratory for Optical and Computational Instrumentation, Wisconsin, USA) software. The image stack was converted to 8-bit, otherwise no manipulation was done. As thrombus cannot be intuitively distinguished in Image J, the identical frame used to measure intensity by Ilumien Optis software was used with the corresponding thrombus structure. The Image was calibrated using the OCT catheter circumference. Intensity was measured using 1) a small rectangular ROI (Region of interest), 2) a large rectangular ROI, and 3) a freehand representative ROI. Luminal (0 mm) ROI was placed as close to the luminal edge as possible. A distance of 0.25 mm in a straight line from center of catheter was measured for the distal (0.25 mm) ROI. A ratio of lightloss through the thrombus was then calculated based on these measurements.

## Histology

Formalin-fixed aspirates were embedded in paraffin, cut in 5 μm sections and stained with Haematoxylin and Eosin (HE) and elastic van Gieson (EvG) stains, respectively, for conventional histomorphological evaluation. The adjacent sections were used for immunohistochemical staining with anti-α smooth muscle actin (SMA, clone 1A4, Dako, GLostrup, Denmark) and anti-CD31 for smooth muscle cells (SMC) and endothelial cells, respectively, to visualize the organization of thrombus.

## Analysis of tissue sections

Sample size of aspirated materials were measured morphometrically on the total tissue area of HE-stained sections in $mm^2$. For histological composition, the presence of thrombus material, plaque material (such as lipid rich debris and/ or foam cells) or both in each aspirate were

recorded. For thrombus composition, we discriminated between red thrombus part (erythrocyte rich in combination with fibrin and/or granulocytes) and white thrombus part (platelet aggregates in combination with fibrin and/or granulocytes). A 'red thrombus' was defined as erythrocyte content of ≥ 70%, a 'white thrombus' was defined as platelet content of ≥ 70%, and a 'mixed thrombus' was defined as both erythrocyte and platelet content < 70%. Thrombus age was determined analytically according to previously published definitions of thrombus age [5] as:1) fresh (up to 1day), composed of layered patterns of morphologically intact platelets, fibrin, erythrocytes and granulocytes; 2) lytic thrombus (1–5 days), characterized by areas of colliquation (lytic) necrosis and/or karyorrhexis granulocytes; and 3) organized thrombus (> 5 days), marked by an ingrowth of SMCs, with or without depositions of connective tissue and capillary vessel ingrowth.

## Statistical analysis

IBM SPSS 24© (IBM, New York, USA) was used for all calculations. Chi Square test was used for comparing thrombus age and colour. For comparison of ratios by measured by different software, we used Pearson's correlations coefficient. For comparing intensity in different colour thrombus, we used One-Way ANOVA. For comparing fresh, lytic and organized elements, we used independent sample T-test. Linear regression was used for predicting erythrocyte and platelet content. For ischemic time we used a two tailed non-parametric test to compare means.

## Interobserver variability

The two investigators were in full agreement on the categorical variable red or white thrombus using a ratio of 0.5 as cut off value.

## Results

### Thrombus colour and age

66 patients (Table 1) were included in the study, of which 52 had successful aspirates of thrombus which were analyzed histologically. During image analysis, definite thrombus was found in 65 patients, but only 42 matched pairs were available in which thrombus could be analyzed by both histology and imaging (Fig 1). Of these, 36 had elements of fresh thrombi, 7 had elements of lytic stage and 8 had elements of an organized thrombus. Looking at erythrocyte and platelet content, 11 were classified as red, 21 as white and 10 as mixed. Out of the fresh thrombi 16 (44%) were classified as white and 10 (28%) as red (p = 0.27), of the lytic thrombi 2 (29%) were classified as white and 3 (43%) were classified as red (p = 0.46) and of the organized thrombi 5 (63%) was classified as white and 2 (25%) were classified as red (p = 0.62).

### Ischemic time

Ischemic time, defined as time from debut to reperfusion, was 200 (±28) min for white, 188 (±27) min for mixed and 167 (±21) min for red thrombi (p = 0.66). Thrombi characterized as fresh had shorter ischemic time than not-fresh; 179 (±14) min vs 249 (±79) min (p = 0.39).

### Intensity ratio by Illumien Optis and ImageJ software

Measured intensity ratio by small ROI (N = 65) and freehand ROI (N = 66) by Image J were highly correlated (0.91, p <0,0001) and slightly less with large ROI (N = 43) (0.80, p <0.0001). Intensity ratio measured by Ilumien Optis software was weakly correlated to ImageJ small ROI measurements (0.40; p = 0.001). Further statistics were therefore performed with both Ilumien Optis measurements and small ROI ImageJ measurements (Fig 2).

**Table 1.**

| Table (N = 66) | | | |
|---|---|---|---|
| Age mean(SD) | | years | 61(10.5) |
| Sex (female) | | % | 26 |
| Body Mass Index mean(SD) | | kg/m$^2$ | 28.2(6.4) |
| Hypertension | | % | 29 |
| Diabetes Mellitus | | % | 8 |
| Statin use | | % | 19 |
| Smoking | | % | 42 |
| Previous PCI | | % | 2 |
| BP systolic mean(SD) | | mmHg | 142(25) |
| BP diastolic mean(SD) | | mmHg | 87(19) |
| Ischemic time mean(SD) | | min | 194(109) |
| Procedure time mean(SD) | | min | 59(24) |
| X-ray contrast volume mean(SD) | | ml | 182(58) |
| Acetylisalicylic acid | | % | 100 |
| Ticagrelor | | % | 48.5 |
| Clopidrogel | | | 51.5 |
| GpIIb/IIIa inhibitor | | % | 68 |
| Culprit vessel | LAD | % | 41 |
| | RCA | % | 42 |
| | CX | % | 14 |
| | Other | % | 3 |
| OCT runs (SD) | | | 1.3(0.7) |

SD = Standard deviation, PCI = Percutaneous Coronary Intervention, BP = Blood Pressure, GPIIb/
IIIa = Glycoprotein receptor IIb/IIIa, LAD = Left Anterior Descending artery, RCA = Right Coronary Artery,
CX = Circumflex artery

## Erythrocyte content

The histological categories red, white and mixed thrombus showed no correlation with intensity measurements by Ilumien Optis (p = 0.41) or ImageJ Small ROI (p = 0,48) by One-Way ANOVA. Intensity ratio could not predict the content of erythrocytes (Standardized Beta -0.014, p = 0.93) or platelets (Beta 0.002, p = 0.99) (Ilumien) (Fig 3).

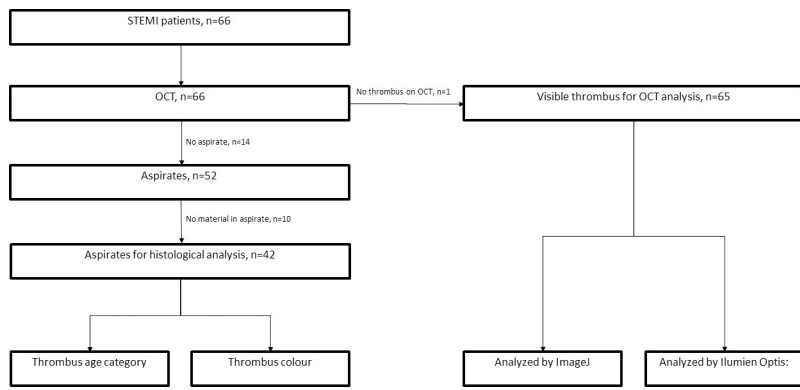

**Fig 1. Flowchart.**

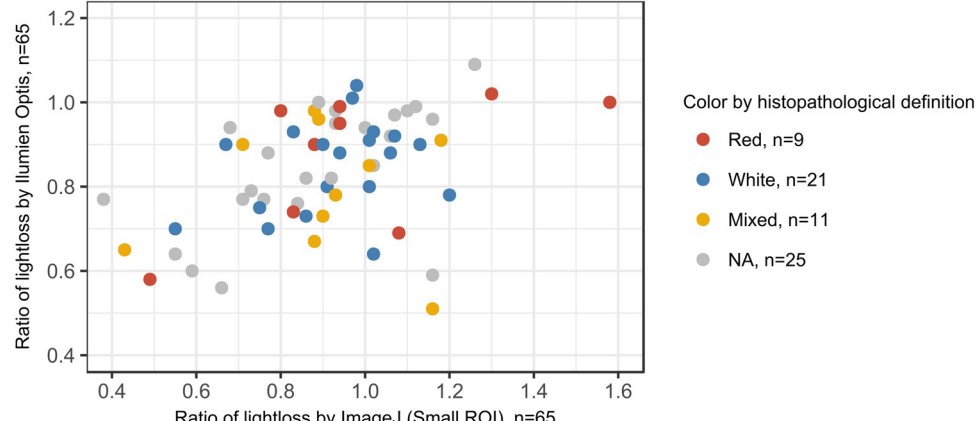

**Fig 2. Comparison between ratio of lightloss measured by Ilumien Optis and ImageJ softwares with correlation 0,40 (p = 0,0001).**

Thrombus age was analyzed by independent sample T-Test, and there were no significant difference in intensity between fresh and non-fresh by Ilumien (p = 0.97) or ImageJ (p = 0.49), between organized and non-organized by Ilumien (p = 0.72) and ImageJ (p = 0.72) or lytic and non-lytic by Ilumien (p = 0.31) and ImageJ (p = 0.63) (Fig 4).

## Discussion

We did not find any correlation between intensity ratio measured by OCT and thrombus content of red blood cells or platelets. The variances in colour between the different histological stages were not significant. Furthermore, OCT could not distinguish between the different

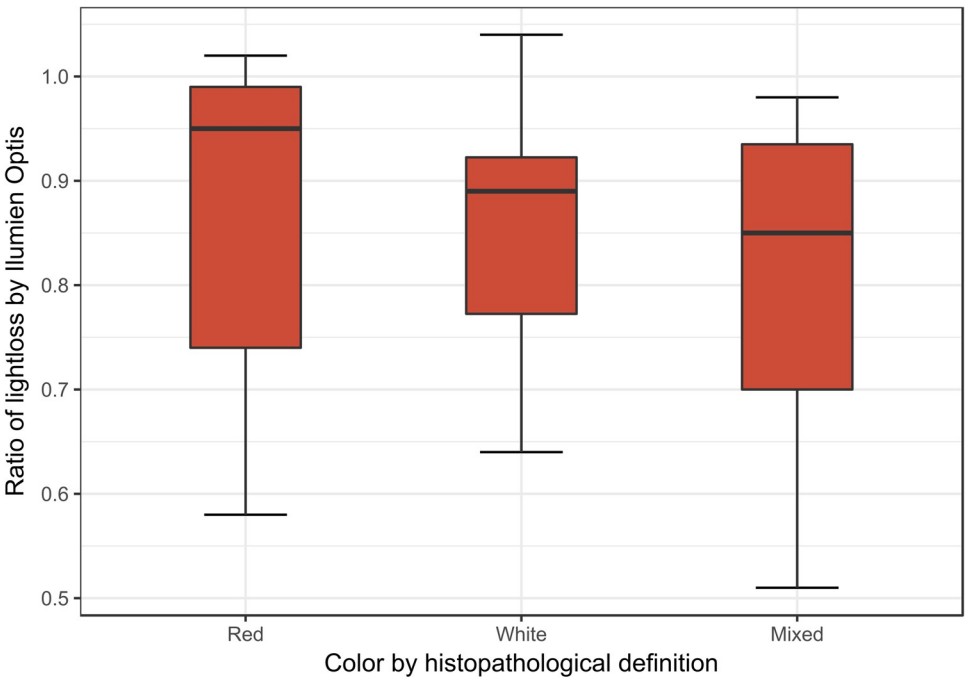

**Fig 3. Ratio of lightloss by Ilumien Optis software in the different histopathological defined colours.**

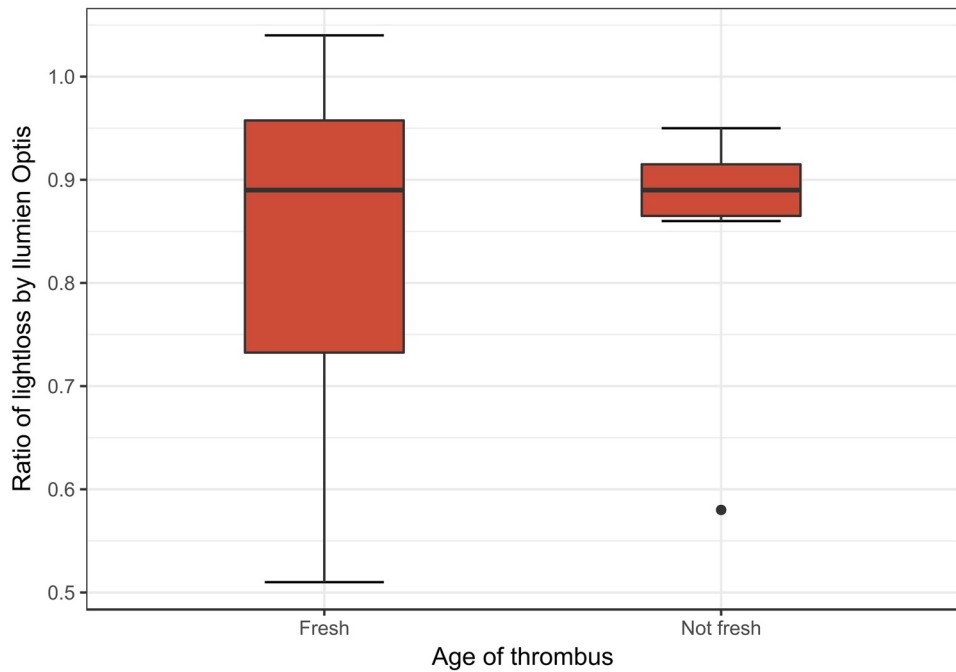

**Fig 4. Ratio of lightloss by Ilumien Optis software in thrombi defined by histopathological appearance.**

histologically defined thrombus stages: fresh, lytic or organized. Fresh thrombi tended to have a shorter ischemic time, though not significant. Paradoxically, ischemic time was shortest for red thrombi. This difference was also not significant.

Our findings contradicts previous workers reporting that red and white thrombi have different appearance on OCT images [6]. The presence of larger thrombi can be established with a conventional coronary angiogram. Using intra vascular ultrasound (IVUS), even smaller volumes of thrombi can be detected. However, only OCT gives images with sufficiently high resolution to potentially evaluate the content of each individual thrombus [9].

OCT uses infrared light and measure the echo time delay from nearby structures giving a distance, and signal intensity, which reveals the optical properties of the structures. During acquisition, the catheter spins around collecting data in a circular or tomographic pattern. This means that structures that are behind objects impenetrable to infrared light will not be visible. Red blood cells are impenetrable to light, meaning that blood must be cleared from the lumen by injecting contrast. Thrombi containing a high proportion of red blood cells will be less penetrable to light than those with less red blood cells. Kume et al [7] used this principle to show that in a macroscopic red thrombus the light intensity decreased more rapidly than in a macroscopic white thrombus with increasing distance from the catheter. A cut of value of > 50% intensity loss over a distance of 0.25 mm was found to correlate with a macroscopic red thrombus with a sensitivity of 90% and specificity of 88%. Our study has a different design than Kume et al. We characterized thrombus composition by histology. In our view, histological examination is a more direct way to determine thrombus composition. Kume et al studied the exact same thrombus sample by OCT and histology, however their OCT study was performed ex vivo on 40 human cadavers and compared to histologic examination post mortem. In our clinical study, OCT was performed in vivo and we compared thrombus remnants in the patient with the histology of aspirated materials. The age and composition of the abluminal parts of the thrombi left behind after aspiration could potentially be more white and platelet

rich, and the luminal aspirated part more red and erythrocyte/fibrin rich. As 36 out of 42 aspirates contained elements of fresh thrombus and 21 out of 42 where classified as white, we at least know that the luminal parts of the thrombus are mostly fresh and to a large extent white. The remnants were only studied by imaging, and were as defined by OCT exclusively red. If the first formed abluminal part of the thrombi matures after it is formed, this would explain that the remnants are all red, and could explain the discrepancy with the ex vivo study. However, it seems unlikely as Silvain et al showed that fibrin content increases with ischemic time in aspirated thrombi [8]. Previous aspiration studies have also shown that the formation of thrombus is more complex with a layered appearance, rather than a continuous gradient of platelet rich to erythrocyte/fibrin rich gradient from the vessel wall to the lumen [5].

OCT has important limitations. Because data are collected in a tomographic pattern, the amount of echoed information will greatly decrease as the distance from the light source increases. This means that the backscattered data from structures close to the catheter will produce more detailed pictures than those further away. The ratio between two fixed points will also be influenced. The Optis Ilumien software compensates for this with by using an algorithm to fill in the missing information [10]. The data are also logarithmically transformed. Although we sought to evaluate if this on-line system could be used clinically, we also analyzed the raw data using the off-line system ImageJ. Kume et al used NIH Image© (National Institute of Health, Wisconsin, USA), a previous version of ImageJ. Choosing size of ROI poses a challenge using ImageJ software. A fixed sample size will cover a much larger anatomical structure further away from the catheter than close to it. The dilemma is therefore whether to include the same number of pixels or an anatomical area of the same size. Structures close to the catheter will appear more compressed (Figs 5 and 6), making it more difficult to identify anatomical structures. Also, a large geometrical ROI, such as a large rectangle, will not fit the irregular structure of a thrombus. We therefore chose to identify thrombi with Ilumien Optis and use the same frame for measurements with ImageJ. We used three different ROIs; two different sizes of fixed rectangular ROI, and one freehand ROI to trace the irregular surface of the

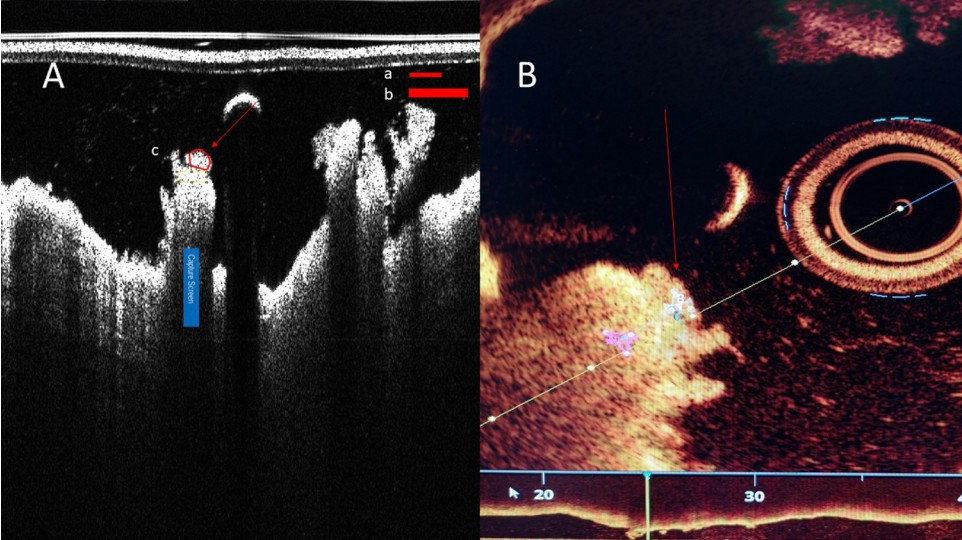

**Fig 5.** The same thrombus (arrow) as it appears on ImageJ (A) and Ilumien Optis (B) softwares. Thrombus protruding towards catheter appears compressed in ImageJ (A). Size of ROI (Region of Interest) showed in red figures (a = small ROI, b = large ROI, c = freehand ROI). Ratio of lightloss in these images was 1.02 by ImageJ and 0.64 by Ilumien Optis. Diameter of catheter is 0.914 microns.

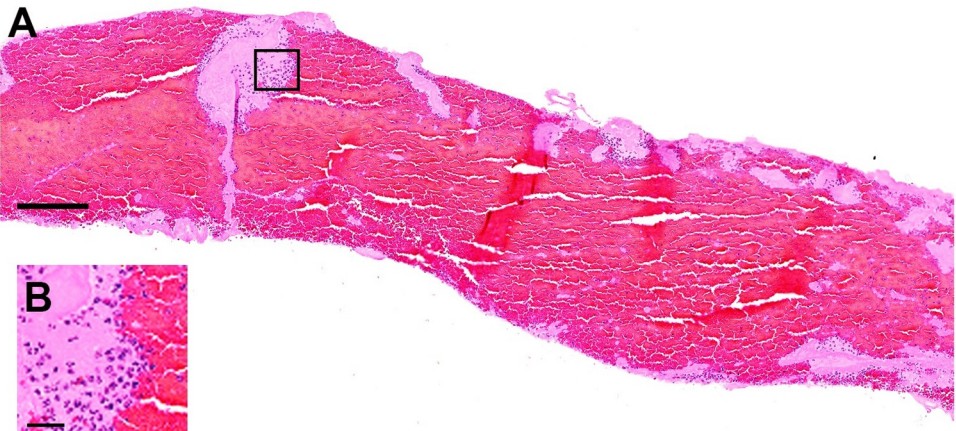

**Fig 6.** Representative image of fresh thrombus, classified as a red thrombus (in this case 80% of erythrocytes) in HE staining (A), red area represents erythrocytes; Inset: a higher magnification image (B) showing the interface of small area of platelet aggregates (pale pink staining) with red blood cells and granulocytes (dark purple nucleated cells). Images are from the same patient as Fig 5. Bar scale in A: 200 μm and in B: 25 μm.

thrombus. The size of small rectangular ROI was chosen so that all images could be analyzed. Using large rectangular ROI, 23 out of 66 images could not be analyzed. All the three different ROIs were highly correlated, so we compared only the fixed small rectangular ROI with measurements from Ilumien Optis.

Our histopathological analysis used a set-up validated by other thrombus aspirations studies at AMC [4, 5]. Rudolf Virchow (1821–1902) first described the mechanism of thrombus formation (150 years ago), which became later known as "Virchow Triad". He divided thrombi into two types: arterial (white) thrombi consisting of platelet aggregates and venous (red thrombi) consisting of mainly red blood cells and fibrin. His theory was that arterial thrombi first forms through activation of platelets, and thus platelet aggregation, leading to stasis and fibrin formation with trapping of erythrocytes transforming the thrombus to red in colour over time. This would mean that red thrombi tend to be older than white thrombi. The time-span in which red thrombus is formed is obviously highly variable, because it depends on diverse factors that evoke significant stasis of blood at the site of the culprit vessel. This theory is supported by modern evidence. Ischemic time is positively correlated with fibrin content, and negatively with platelet content [8, 11]. From clinical experience we know that thrombi behaves very differently in the individual STEMI patient. Problems with high thrombus burden, distal embolization and no-flow situations are well known. If the age and content of a thrombus could be determined peroperatively, this could give the operator a better understanding of the thrombus at hand.

The age of a thrombus can be classified reliably according to standardized histopathological appearance of tissue decay followed by repair in the aspired materials. The presence of older age thrombus has been showed to correlate to higher mortality [5, 12]. The mechanism is uncertain. A longer period of thrombus formation and lysis could result in a prolonged period of ischaemic myocardial damage. A process of clotting and lysis could result in more distal embolization, which would compromise the microvascular bed of the myocardium. The increased mortality could also be a result of patient confounding factors, such as sensitivity to chest pain and at what urgency the patients seeks medical help. We know presently that the formation and fate of thrombi are much more complex than Virchow's theory. Smooth muscle cells and neutrophil granulocytes play a specific and complex role, with expression of cytokines

such as matrix metalloproteinase (MMP) and creating neutrophil extracellular traps (NETs). The expression of MMPs and the amount of NETs varies with thrombus age [12–14]. As well as promoting thrombogenicity, this could also affect the healing of the implanted stent [15–19]. Finding an optimal treatment strategy is therefore more complex than what we can hope to visualize with intracoronary imaging alone.

## Patient selection and generalizability of results

We chose to study thrombi from STEMI patients. This is a relatively homogenous group presenting with similar clinical picture and with a clear ECG indication for intervention. In most cases the presence of coronary thrombi can be assumed, and thrombus aspiration was, at the time of submitting the protocol, routine practice [20]. Our cohort represents a selection of STEMI patients. Patients with complex lesions, tortuous anatomy and heavy calcifications were excluded. Our findings cannot be generalized without some considerations. However, the selection was done in order to be able to aspirate thrombus and to get the best quality OCT images. We do believe that including all STEMI patients would cloud the data with poorer quality of OCT images and fewer thrombi aspirated. It is also an investigation of methodology, transferring findings in a standardized laboratory setting to an acute clinical scenario.

## Aspiration method

Our method for thrombus aspiration was using an off the shelf aspiration catheter. Export Advance catheter was chosen as this comes with a 40 micron filter basket. Other methods for extracting thrombus and for intracoronary imaging such as larger mother-and-child catheter or angioscopy are not feasible in this population due to the acute setting.

## Medication

All patients received loading dose of acetylic salisylic acid and either ticagrelor or clopidrogel before angiography. All were given heparin at dose of 5000 IU to 7500 IU at the start of the procedure. This reflects clinical practice, but could alter the coagulation state and thrombus appearance. None were given GpIIb/IIIa inhibitors before thrombus aspiration and OCT.

## Limitations

Our study is limited by having a small number of subjects, and limited number of paired subjects with both histological and imaging. The thrombus aspiration was performed before imaging, so the aspirated content might be different from the remnants analyzed by OCT

## Conclusion

The consensus document for OCT measurements states that thrombus composition can be predicted by measuring light intensity in thrombus relying on only one reference article using post mortem thrombus analysis. We have not been able to reproduce these findings in a clinical setting, either by use of online llumien Optis software or the offline software ImageJ. We have not found a relationship between erythrocyte content and thrombus age. In our study OCT could not predict whether the thrombus contains fresh, organized or lytic elements. One reason for this could be the limitations in the OCT system itself.

## Supporting information

**S1 File. Supporting data.**
(DAT)

**S1 Text.**
(DOCX)

## Acknowledgments

Special acknowledgments to Niels Ramsing Holm and Omeed Neghabat at the OCT core lab at Aarhus University Hospital for help with OCT analysis, and to Kjetil Løland and Cedric Davidsen at Haukeland University Hospital for help statistics and figures.

## Author Contributions

**Conceptualization:** Erlend Eriksen, Vegard Tuseth, Jan Erik Nordrehaug, Øyvind Bleie, Allard C. van der Wal.

**Data curation:** Erlend Eriksen.

**Formal analysis:** Erlend Eriksen, Jon Herstad, Øyvind Bleie, Allard C. van der Wal.

**Funding acquisition:** Erlend Eriksen, Jan Erik Nordrehaug, Øyvind Bleie.

**Investigation:** Erlend Eriksen, Jon Herstad, Øyvind Bleie, Allard C. van der Wal.

**Methodology:** Erlend Eriksen, Jon Herstad, Øyvind Bleie, Allard C. van der Wal.

**Project administration:** Erlend Eriksen.

**Resources:** Erlend Eriksen.

**Software:** Erlend Eriksen.

**Supervision:** Øyvind Bleie, Allard C. van der Wal.

**Validation:** Erlend Eriksen, Jon Herstad.

**Writing – original draft:** Erlend Eriksen.

**Writing – review & editing:** Jon Herstad, Kartika Ratna Pertiwi, Vegard Tuseth, Jan Erik Nordrehaug, Øyvind Bleie, Allard C. van der Wal.

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
