## [Decision Letter · Decision Letter 0]

20 Jan 2022

PONE-D-21-35338Thrombus characteristics evaluated by acute Optical Coherence Tomography in ST Elevation Myocardial InfarctionPLOS ONE

Dear Dr. Eriksen,

Thank you for submitting your manuscript to PLOS ONE. After careful consideration, we feel that it has merit but does not fully meet PLOS ONE’s publication criteria as it currently stands. Therefore, we invite you to submit a revised version of the manuscript that addresses the points raised during the review process.

The reviewers were generally favorable toward your work, but still highlighted some concerns and areas for improvement within the manuscript. Please address each concern individually in the response letter, and we thank you for your submission.

We look forward to receiving your revised manuscript.

Kind regards,

R. Jay Widmer

Academic Editor

PLOS ONE

Journal Requirements:

2. Please remove the CONSORT flowchart and checklist, and the clinical trial protocol. These documents were erroneously requested upon submission since you selected 'clinical trial' as article type. However, we realize that your study is not directly related to the original clinical trial and therefore we processed it as a Research Article (where the clinical trial documentation is not needed). We apologize for the inconvenience.

In your manuscript, please clarify whether your original clinical trial has been published, providing a reference if applicable.

Thank you for your attention to these requests.

Reviewers' comments:

Reviewer's Responses to Questions

**Comments to the Author**

1. Is the manuscript technically sound, and do the data support the conclusions?

Reviewer #1: Yes

Reviewer #2: Yes

2. Has the statistical analysis been performed appropriately and rigorously? 

Reviewer #1: Yes

Reviewer #2: Yes

3. Have the authors made all data underlying the findings in their manuscript fully available?

Reviewer #1: Yes

Reviewer #2: Yes

4. Is the manuscript presented in an intelligible fashion and written in standard English?

Reviewer #1: Yes

Reviewer #2: Yes

5. Review Comments to the Author

Reviewer #1: The authors sought to compare erythrocyte content and age of intracoronary thrombus assessed by histopathology and optical coherence tomography (OCT). The main conclusion of this study is that OCT could not differentiate between red and white thrombi, nor determine thrombus age.

Major concerns are:

1. As for such type of research, sample size is small.

2. The composition of the intracoronary thrombus depends on a distance from the vessel wall. Thrombi contain more fibrin in the superficial part, near the vessel wall whereas in the core they contain more erythrocytes. Thus, histopathology and OCT assessing not necessary the same part of thrombi were less likely to achieve a convergent result. This issue requires thorough discussion.

3. Data regarding relationships between time of ischemia and thrombus composition in this sample size could be provided.

4. Independent determinants of histopathological content of thrombi versus OCT lightloss through the thrombus allow to compare their causative mechanisms

Reviewer #2: Dear Authors,

Your great clinical work is highly appreciated. Meticulously performed, great analysis.

Some major factor missing: time of chest pain (or ECG-diagnosis) to time of successful wiring. You should correlate it with the OCT and histology findings.

Small issue: please, describe the details of OCT imaging (ml/s contrast, manual trigger?, how many runs were needed?).

Your control of OCT automatic assessment with ImageJ is an outstanding concept.

Writing, typo, figures, referencing perfect.

However, I would not stress your "message" that these results might lead to change to current (pharmacologic? interventional?) approach to STEMI. Similarly, in the abstract, I would not this sentence: OCT to tailor anti-thrombotic treatment".

I reckon that you (correctly) excluded patients with heavy calcification and complex lesions. I also disagree your opinion in the Discussion that you "have no reason to believe that our results would be different in more complex disease". Please give explanation.

Sincerely yours

6. PLOS authors have the option to publish the peer review history of their article (what does this mean?). If published, this will include your full peer review and any attached files.

Reviewer #1: No

Reviewer #2: No

---

## [Author Response · Author response to Decision Letter 0]

15 Feb 2022

Dear Reviewers, 

Below are my rebutting comments. I hope the issues are addressed satisfactory. 

The numbers in parantheses refers to line number in Manuscript with tracker changes. 

Reviewers' comments:

Reviewer's Responses to Questions

Comments to the Author

1. Is the manuscript technically sound, and do the data support the conclusions?

Reviewer #1: Yes

Reviewer #2: Yes

2. Has the statistical analysis been performed appropriately and rigorously?

Reviewer #1: Yes

Reviewer #2: Yes

3. Have the authors made all data underlying the findings in their manuscript fully available?

Reviewer #1: Yes

Reviewer #2: Yes

4. Is the manuscript presented in an intelligible fashion and written in standard English?

Reviewer #1: Yes

Reviewer #2: Yes

5. Review Comments to the Author

Reviewer #1: The authors sought to compare erythrocyte content and age of intracoronary thrombus assessed by histopathology and optical coherence tomography (OCT). The main conclusion of this study is that OCT could not differentiate between red and white thrombi, nor determine thrombus age.

Major concerns are:

1. As for such type of research, sample size is small.

I agree that the sample size is small. Unfortunately we had to terminate inclusion early, due to safety concerns of the Absorb© bioresorbable scaffold. However our reference studies have similar sample sizes. Kumar et al included 40 human cadavers and Silvain et al studied 45 thrombi. We think our study increases the knowledge about thrombi in ST elevation myocardial infarct, although it is not definite evidence. 

2. The composition of the intracoronary thrombus depends on a distance from the vessel wall. Thrombi contain more fibrin in the superficial part, near the vessel wall whereas in the core they contain more erythrocytes. Thus, histopathology and OCT assessing not necessary the same part of thrombi were less likely to achieve a convergent result. This issue requires thorough discussion.

It is a good point that the remnants might be different from the aspirates, which we have mentioned in the discussion. We have elaborated on the issue under ‘Discussion’ (318-328). 

3. Data regarding relationships between time of ischemia and thrombus composition in this sample size could be provided.

Ischemic time is found in Table. A more thorough analysis of subgroups has been added in ‘Results’ (259-263)

4. Independent determinants of histopathological content of thrombi versus OCT lightloss through the thrombus allow to compare their causative mechanisms

I understand this question as a request for analysis of determinants of thrombus contents in the vessesl wall, such as type of plaque, presence of lipid, calcification, thin cap atheroma or erosion. Unfortunatly most of the vessel wall is obscured by shadow from the thrombus, so that the characteristics of the vessel wall cannot be distinguished. 

Reviewer #2: Dear Authors,

Your great clinical work is highly appreciated. Meticulously performed, great analysis.

Some major factor missing: time of chest pain (or ECG-diagnosis) to time of successful wiring. 

You should correlate it with the OCT and histology findings.

Ischemic time is found in Table. A more thorough analysis of subgroups has been added in ‘Results’ (256-259)

Small issue: please, describe the details of OCT imaging (ml/s contrast, manual trigger?, how many runs were needed?).

The details has been elaborated in ‘Methods’ (162-164). Number of runs has been added to Table. 

Your control of OCT automatic assessment with ImageJ is an outstanding concept.

Writing, typo, figures, referencing perfect.

However, I would not stress your "message" that these results might lead to change to current (pharmacologic? interventional?) approach to STEMI. Similarly, in the abstract, I would not this sentence: OCT to tailor anti-thrombotic treatment".

I agree that this message might be exaggerated. It was an attempt to relate the aim of the study to everyday clinical practice. The statement has been modified (92-93, 373-377 and 393-396). 

I reckon that you (correctly) excluded patients with heavy calcification and complex lesions. I also disagree your opinion in the Discussion that you "have no reason to believe that our results would be different in more complex disease". Please give explanation.

Sincerely yours

I agree that we have no evidence showing thrombus formation behaves similarly in more complex and calcified lesions. However since our main finding is that OCT cannot differentiate thrombus content, we have no reason to believe that OCT is a better tool in such cases. Nevertheless, the statement has been cancelled (408-409). 

Kind Regards, 

Erlend Eriksen

---

## [Decision Letter · Decision Letter 1]

24 Mar 2022

Thrombus characteristics evaluated by acute optical coherence tomography in ST elevation myocardial infarction

PONE-D-21-35338R1

Dear Dr. Eriksen,

We’re pleased to inform you that your manuscript has been judged scientifically suitable for publication and will be formally accepted for publication once it meets all outstanding technical requirements.

Kind regards,

R. Jay Widmer

Academic Editor

PLOS ONE

Reviewers' comments:

Reviewer's Responses to Questions

**Comments to the Author**

1. If the authors have adequately addressed your comments raised in a previous round of review and you feel that this manuscript is now acceptable for publication, you may indicate that here to bypass the “Comments to the Author” section, enter your conflict of interest statement in the “Confidential to Editor” section, and submit your "Accept" recommendation.

Reviewer #1: All comments have been addressed

Reviewer #2: All comments have been addressed

2. Is the manuscript technically sound, and do the data support the conclusions?

Reviewer #1: Yes

Reviewer #2: Yes

3. Has the statistical analysis been performed appropriately and rigorously? 

Reviewer #1: Yes

Reviewer #2: Yes

4. Have the authors made all data underlying the findings in their manuscript fully available?

Reviewer #1: Yes

Reviewer #2: Yes

5. Is the manuscript presented in an intelligible fashion and written in standard English?

Reviewer #1: Yes

Reviewer #2: Yes

6. Review Comments to the Author

Reviewer #1: My all comments have been addressed in an adequate way therefore I have no more comments. I suggest accept.

Reviewer #2: (No Response)

7. PLOS authors have the option to publish the peer review history of their article (what does this mean?). If published, this will include your full peer review and any attached files.

Reviewer #1: No

Reviewer #2: No

---

## [Editor Report · Acceptance letter]

1 Apr 2022

PONE-D-21-35338R1 

Thrombus characteristics evaluated by acute optical coherence tomography in ST elevation myocardial Infarction 

Dear Dr. Eriksen:

I'm pleased to inform you that your manuscript has been deemed suitable for publication in PLOS ONE. Congratulations! Your manuscript is now with our production department. 

Kind regards, 

on behalf of

Dr. R. Jay Widmer 

Academic Editor

PLOS ONE